# Visually Grounded Continual Language Learning with Selective Specialization

**Kyra Ahrens    Lennart Bengtson    Jae Hee Lee    Stefan Wermter**

University of Hamburg

{kyra.ahrens,lennart.bengtson,jae.hee.lee,stefan.wermter}@uni-hamburg.de

## Abstract

A desirable trait of an artificial agent acting in the visual world is to continually learn a sequence of language-informed tasks while striking a balance between sufficiently specializing in each task and building a generalized knowledge for transfer. *Selective specialization*, i.e., a careful selection of model components to specialize in each task, is a strategy to provide control over this trade-off. However, the design of selection strategies requires insights on the role of each model component in learning rather specialized or generalizable representations, which poses a gap in current research. Thus, our aim with this work is to provide an extensive analysis of selection strategies for visually grounded continual language learning. Due to the lack of suitable benchmarks for this purpose, we introduce two novel diagnostic datasets that provide enough control and flexibility for a thorough model analysis. We assess various heuristics for module specialization strategies as well as quantifiable measures for two different types of model architectures. Finally, we design conceptually simple approaches based on our analysis that outperform common continual learning baselines.[1] Our results demonstrate the need for further efforts towards better aligning continual learning algorithms with the learning behaviors of individual model parts.

## 1 Introduction

Grounding language in visual perception is a crucial step towards agents that effectively understand and interact with the physical world (Bisk et al., 2020). In a realistic setting, such an agent would require the ability to continuously integrate novel experience and skills with existing knowledge in an open-ended process, a challenge commonly known as *Continual Learning* (CL) (Chen and Liu, 2018; Parisi et al., 2019; Wang et al., 2023).

---

[1]Code and datasets will be made available at https://github.com/ky-ah/selective-lilac.

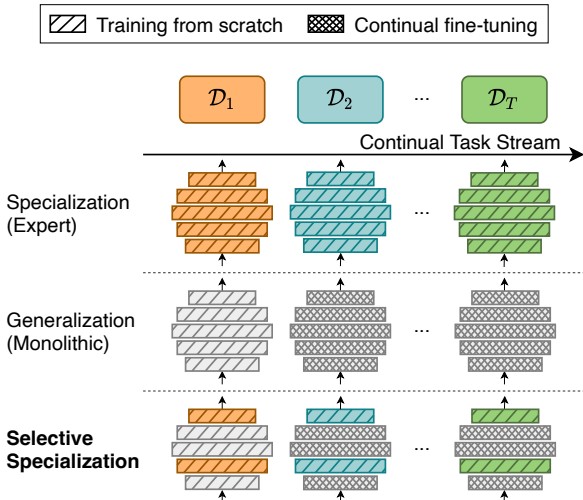

Figure 1: The specialization-generalization trade-off in a continual learning setting. *Expert* solutions train a copy of the network for each new task, while *monolithic* networks make unrestricted updates to all parameters. In this paper, we focus on *selective specialization* to strike a balance between full specialization and generalization.

Ideally, the underlying neural model of an agent would sufficiently solve each isolated task (i.e., *specialization*) while harnessing the shared structure and subproblems underlying all tasks to create generalizable representations for knowledge transfer (i.e., *generalization*). A strategy to provide control over this trade-off is to introduce task-specific parameters to a carefully selected subset of model components (i.e., *selective specialization*), as can be seen in Fig. 1.

Understanding which model components (here referred to as *modules*) are suited for task-specialization requires insights on their role in solving each task, which prior works on continual Vision-Language (VL) grounding fail to explain. At the same time, there is a lack of suitable benchmarks that allow for a fine-grained model analysis, as existing CL scenarios for language grounding based on synthetic images are too simplistic concerning the CL problem (e.g., only one

distributional shift) (Greco et al., 2019) or the VL grounding problem (e.g., single-object, trivial language) (Skantze and Willemsen, 2022), while those based on real-world images (Srinivasan et al., 2022; Jin et al., 2020) may encourage models to take shortcuts in the vision-language reasoning process, which leaves the generalizability of statements derived from any model analysis questionable. To this end, we introduce the LIfelong LAnguage Compositions (LILAC) benchmark suite that comprises two diagnostic VL datasets that allow for investigating the continual learning behavior of the models with a high degree of control and flexibility while being challenging enough to require object localization, spatial reasoning, concept learning, and language grounding capabilities of the continual learner.

Based on the two proposed LILAC benchmarks, we conduct a fine-grained analysis of two different vision-language model architectures in this work that comprises the following steps: (i) We analyze whether selective specialization generally benefits from dividing the learning process into the two intermittent stages of *task adaptation*, where only the specialized parameters are being updated, and *knowledge consolidation*, where only the shared parameters are updated with respect to the task-specific learned representations. (ii) We derive and evaluate heuristics from prior literature about introducing task-specific parameters with respect to different layer depths and specific VL network modules. (iii) We assess the suitability of parameter importance measures from pruning research as indicators of the efficacy of module selection strategies.

We conclude our work by demonstrating the superior performance of module selection strategies found in our analysis, when trained under the adaptation-consolidation (A&C) procedure as described above, over common CL baselines. Thus, we summarize the contributions made in our work as follows:

1. We propose two novel datasets for visually grounded continual language learning whose problem spaces have a well-defined shared structure, thus inherently promoting a careful selection of specialized modules (cf. Sec. 3).
2. We provide a thorough analysis of the efficacy of different selective specialization strategies for two representative vision-language architectures (cf. Sec. 4.2.1–Sec. 4.2.3).

3. We show that carefully balancing the trade-off between generalization and specialization via selective specialization helps us design simple approaches that outperform CL baselines (cf. Sec. 4.2.4), ultimately demonstrating the importance of aligning CL methods with the learning behavior of individual model components.

## 2 Background

### 2.1 Continual Learning Setting

An overview of the learning setting is provided in Fig. 2. We consider a VL model composed of a language encoder $g(\cdot)$, a visual feature extractor $h(\cdot)$, a VL fusion network $f_\theta(\cdot)$, and a decoder $d(\cdot)$. We initialize the model at time $t_{\text{init}}$ and freeze $g$, $h$, and $d$ afterwards. After initialization, the model learns an ordered sequence of tasks from their training sets $\mathcal{D}_t$ for $t \in \{1, 2, \ldots, T\}$ and updates VL fusion parameters $\theta$ to perform well on the test sets. We assume access to the task identity of every model input during training and testing. Each data point in $\mathcal{D}_t$ is composed of visual observations and input instructions $(\mathbf{l}_t, \mathbf{o}_t, (\mathbf{o}_t^+, \mathbf{o}_t^-))$ where $\mathbf{o}_t^+$ and $\mathbf{o}_t^-$ denote the visual scenes corresponding to a correct and a wrong execution of the instruction $\mathbf{l}_t$ upon observing $\mathbf{o}_t$, respectively. This setting is an extension of Natural Language Inference (Bowman et al., 2015) towards visual language grounding that can be approached using contrastive learning (Li et al., 2023), where we consider $(\mathbf{o}_t, \mathbf{l}_t)$ to be an image-text premise and $\mathbf{o}_t^+, \mathbf{o}_t^-$ to be visual hypotheses.

### 2.2 The Specialization-Generalization Trade-off

Let $\theta_t$ denote the set of parameters that contains a subset of parameters specific to task $t$. Then we can express the learning objective as finding parameter values $\theta_1, \ldots, \theta_T$ for network $f_{\theta_t}$ that maximize the average accuracy across all test sets. From this objective arises a parameter-sharing trade-off between *specialization* and *generalization* that can be addressed by different design choices for $f$ (Ostapenko et al., 2021).

One approach is to use a monolithic network with parameters to be fully shared across tasks such that $\theta = \theta_t \, \forall \, t \in \{1, \ldots, T\}$ (Kirkpatrick et al., 2017; Chaudhry et al., 2019b), which facilitates knowledge transfer between tasks at the cost of forgetting. The opposite approach is to

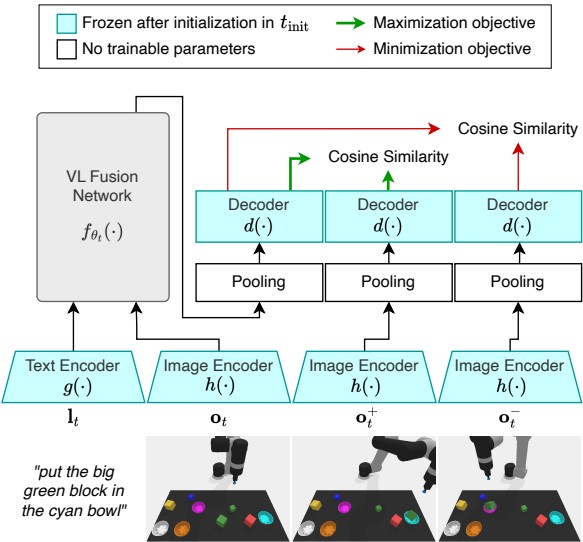

Figure 2: Overview of the model training on a LILAC-3D example. Each of the $L$ vision-language fusion layers (gray box) contains modules $m$ that are candidates for specialization.

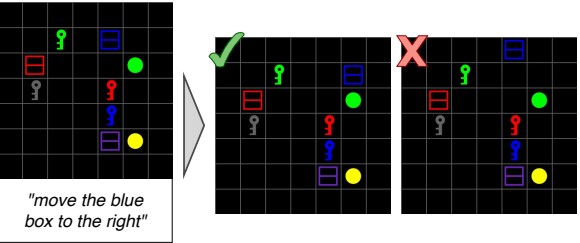

Figure 3: Example of the LILAC-2D dataset. Based on the instruction and the visual premise, a model learns to separate the two visual hypotheses (true target image, false target image) corresponding to a right and a wrong understanding of the instruction, respectively.

train task-specific *expert solutions* (Aljundi et al., 2017; Rusu et al., 2016) such that $\theta_i \cap \theta_j = \emptyset$ for $i, j \in \{1, \ldots, T\}$ with $i \neq j$, which alleviates forgetting at high memory most and impeded transfer. van de Ven and Tolias (2019) propose to balance this trade-off by learning task-specific "multi-headed" output layers. However, this approach assumes a uniform feature space and does not work for VL fusion networks that are trained continually.

Inspired by works on modular continual learning (Ostapenko et al., 2021; Mendez and Eaton, 2021; Mendez et al., 2022), we consider $f$ as a set $\mathcal{M}$ of *modules*, where a module $m$ in this context can be any kind of self-contained parametric function in $f$ (e.g., a convolutional layer or a batch normalization layer).[2] Our aim is to find a strategy for selecting a subset $\mathcal{S} \subset \mathcal{M}$ of modules that is task-specific and parameterized by $\theta_t^{\mathcal{S}}$ for $t \in \{1, \ldots, T\}$, while keeping the remaining modules shared across tasks, thus parameterized by $\theta^{\mathcal{M} \setminus \mathcal{S}}$, yielding $\theta_t = \{\theta_t^{\mathcal{S}}, \theta^{\mathcal{M} \setminus \mathcal{S}}\}$. Notably, $\mathcal{S} = \emptyset$ for a monolithic (or fully shared) network and $\mathcal{S} = \mathcal{M}$ for the expert solutions.

---

[2]Hence, the notion of a module in this paper is more general than that in a neural module network (Andreas et al., 2016).

## 2.3 Intermittent Adaptation and Consolidation (A&C)

We apply a simplified version of the lifelong compositional learning scheme introduced in Mendez and Eaton (2021) that is inspired by Piaget's theories on intellectual development (Piaget, 1976) and has been successfully applied in CL of neural architectures with specialized modules: Instead of updating $\theta_t$ jointly upon learning task $t$, we repeatedly alternate between multiple adaptation steps (here, epochs) to update the task-specific parameters $\theta_t^{\mathcal{S}}$ (assimilation, fast learning) and a single consolidation step that updates the shared parameters $\theta^{\mathcal{M} \setminus \mathcal{S}}$ (accommodation, slow learning). We show in Sec. 4.2.1 that the adaptation-consolidation (A&C) learning scheme can consistently improve performance over joint optimization of $\theta_t$. The algorithm pseudocode can be found in Sec. A.5.

## 3 LILAC Datasets

In what follows, we propose two benchmark datasets that explicitly model the compositional nature of a problem, thus maintaining a high degree of overlap across tasks. This should encourage the models to use as few task-specific parameters as possible and facilitate the exploration of a reasonable selection strategy for specialized modules. During training, the model receives as input three images representing objects in a simulated environment, along with a templated language instruction. Examples of the two proposed datasets can be found in Fig. 2 and Fig. 3.

**LILAC-2D tasks.** This dataset is based on the `minigrid` (Chevalier-Boisvert et al., 2018) environments. For each example, three to nine objects are randomly placed in a $7 \times 7$ grid. Each instruction describes a desired interaction with a desig-

nated target object and takes the form of *"move the <color> <object> <direction>"*, where we choose from a set of six colors, three object types, and four directions. All three subproblems described by the instruction can be orthogonally combined, yielding $6 \times 3 \times 4 = 72$ distinct instructions. False target images are generated by randomly choosing one of the three subproblems to be wrongly solved, e.g., a blue key is moved down, although a green key was supposed to be moved down. The LILAC-2D dataset comprises 500 train, 100 validation, and 100 test samples per instruction, yielding a total of 36,000 train and 7,200 validation and test samples, respectively. For the continual learning stream, we construct $T = 10$ tasks comprising training samples of six different instructions each and keep samples from the remaining 12 instructions for model initialization at time $t_{\text{init}}$.

**LILAC-3D tasks.** To further narrow the gap to real 3D images while maintaining a high degree of control and flexibility for model analysis, we additionally propose LILAC-3D, a dataset with increased spatial complexity and distracting information. It is based on the simulated Ravens (Zeng et al., 2021) benchmark and its extension towards language instructions (Shridhar et al., 2022). Images show a tabletop scenario, where between five and eight blocks and between three and four bowls are randomly placed within the range of a robot arm. Instructions describe a pick-and-place operation with a target block and a target bowl and take the form of *"put the <size> <color1> block in the <color2> bowl"*, where we choose from a set of two different sizes, six block colors, and six bowl colors, thus yielding a total of $2 \times 6 \times 6 = 72$ distinct instructions. The sets of colors for blocks and bowls are fully disjoint to allow the objects to be clearly distinguished from one another. False target images are designed in a way that either the wrong block and the right bowl or the right block and the wrong bowl are chosen for interaction. The dataset statistics as well as the continual stream design are similar to those of the LILAC-2D dataset.

## 4 Evaluations

### 4.1 Experimental Setup

**Model architectures.** We conduct experiments with a transformer encoder (Vaswani et al., 2017) and a feature-wise linear modulation (FiLM) network (Perez et al., 2018) as our VL fusion networks

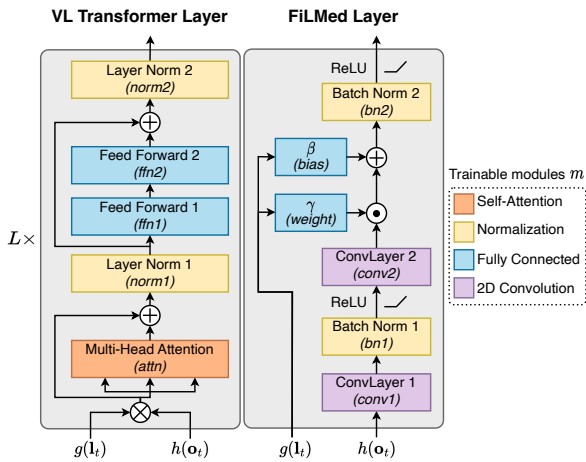

Figure 4: The two VL fusion networks $f_{\theta_t}(\cdot)$ used in our experiments consist of $L$ consecutive VL transformer or FiLMed layers. Each network receives as input the encoded language input and vision input to produce a joint representation $f_{\theta_t}(g(\mathbf{l}_t), h(\mathbf{o}_t))$. Each colored box is a module $m$ and one or multiple modules can be part of a selection strategy $\mathcal{S}$.

$f_{\theta_t}(\cdot)$, which are both well established in research on visual language grounding both in supervised and imitation learning settings (cf. Tan and Bansal, 2019; Chen et al., 2020; Panos et al., 2023; Hui et al., 2020; Lee et al., 2022). An overview of the two VL fusion networks can be found in Fig. 4, where we also indicate the modules that we consider for selection strategies in the experiments. We use the first block of a ResNet-18 (He et al., 2016) as image encoder $h(\cdot)$. For the VL transformer, $g(\cdot)$ is the concatenation of word embeddings of the input instruction, whereas for the FiLMed network, $g(\cdot)$ is a sequence encoding by a single-layer Gated Recurrent Unit (GRU) (Cho et al., 2014). The projection decoder $d(\cdot)$ is a linear fully-connected layer. The VL fusion is followed by a max-pooling operation for FiLM and a mean-pooling operation for the transformer. All model configurations were found based on extensive hyperparameter tuning to achieve maximum performance on each validation set in the i.i.d. training setting and are described in detail in Appendix A.

**Baselines.** We compare to the following baselines: Sequential fine-tuning (**SFT**) performs unrestricted updates on a monolithic architecture with all module parameters shared across tasks and is usually considered as a lower bound for CL model performance. Experience replay (**ER**) (Chaudhry et al., 2019b) is an extension to SFT which stores samples in a fixed-size buffer via reservoir sam-

pling, from which samples are drawn for rehearsal. Elastic Weight Consolidation (**EWC**) (Kirkpatrick et al., 2017) is another extension to SFT that regularizes parameter updates depending on their relative importance. For our experiments, we use the Online EWC version (Schwarz et al., 2018) that does not require storing a separate approximation of the Fisher information matrix per task. Considering a potential upper bound for model performance, multi-task training (**MTL**) optimizes a monolithic architecture on all tasks in the i.i.d. training setting and independent experts (**Expert**) train a randomly initialized set of modules separately for each task.

**Evaluation metrics.** We report the average accuracy across all tasks under a choice of parameters $\theta$ as $\text{ACC}^{(\theta)}$ (or, $\text{ACC}^{(\theta_{1:T})}$ for the set of all task-specific parameters $\theta_{1:T} = \{\theta_1, \ldots, \theta_T\}$). Furthermore, we measure the accuracy gain under a specialization strategy $\mathcal{S}$ compared with using a monolithic network as

$$\Delta\text{ACC}(\mathcal{S}) = \text{ACC}^{(\theta_{1:T})} - \text{ACC}^{(\theta)}, \quad (1)$$

where $\theta_{1:T} = \{\theta^{\mathcal{S}}_{1:T}, \theta^{\mathcal{M}\setminus\mathcal{S}}\}$. Both $\text{ACC}^{(\theta_{1:T})}$ and $\Delta\text{ACC}(\mathcal{S})$ are measured after training on the last task $T$.

## 4.2 Results

In what follows, we first examine whether the A&C learning scheme is useful for training with a module selection strategy $\mathcal{S}$ (cf. Sec. 2.2), such that we can use A&C throughout our experiments (Sec. 4.2.1). Next, we analyze different selection strategies and compare the results with findings from prior literature (Sec. 4.2.2). We then assess whether the efficacy of selection strategies can be quantified using importance scores from pruning research (Sec. 4.2.3). Finally, we compare several selection strategies with CL baselines (Sec. 4.2.4). Each experiment is run ten times with different random seeds that affect parameter initialization and task order in the continual stream.

### 4.2.1 Does the A&C learning scheme benefit training with a module selection strategy?

To assess the effectiveness of the A&C learning scheme, we measure the difference in accuracy gain under the selection strategy $\mathcal{S} = \{m\}$ for each module $m \in \mathcal{M}$, as shown in Fig. 5. Overall, we observe a positive effect of introducing A&C for the majority of specialized FiLM modules and

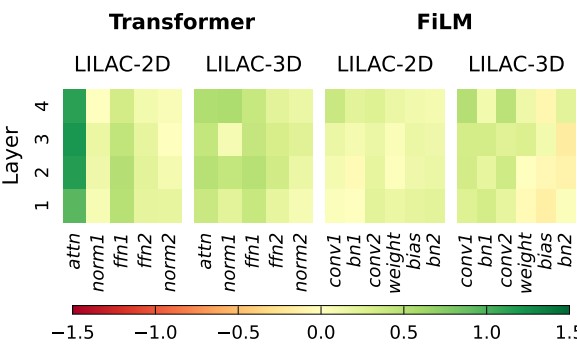

Figure 5: Difference between accuracy gains from isolating each module between A&C (cf. Alg. 1) and joint training of all shared and task-specific modules. Green values indicate the superiority of A&C over joint training.

even consistently across all VL transformer modules. This indicates that despite being updated less frequently, the shared network modules learn the subproblem underlying all tasks sufficiently well while being less prone to forgetting.

### 4.2.2 Does the performance of different module selection strategies align with insights about general model behavior from prior literature?

**Specialization at different layer depths.** To analyze the effect of isolating task parameters at different layer depths for specialization, we construct one model for each of the transformer and FiLMed layers to be specialized, respectively. The results can be found in the first column of Fig. 6.

For the VL transformer, we observe that specialization of late layers yields a higher accuracy gain than specialization of early layers. This indicates that parameters of early transformer layers benefit from being shared across tasks, as such layers learn transferable representations that should be slowly learned via infrequent updates. Such findings confirm prior research on transformers for natural language processing (Hao et al., 2019; Tenney et al., 2019) claiming that late layers capture most local syntactic phenomena, while early layers capture more general semantics.

In their analysis of FiLMed networks, Perez et al. (2018) claim that early layers perform low-level reasoning, such as querying attributes of an object, while late layers perform high-level reasoning, such as comparing two objects. As solving the LILAC-2D tasks requires the model to identify and interact with a single object, whereas for solving the LILAC-3D tasks, two objects would have to

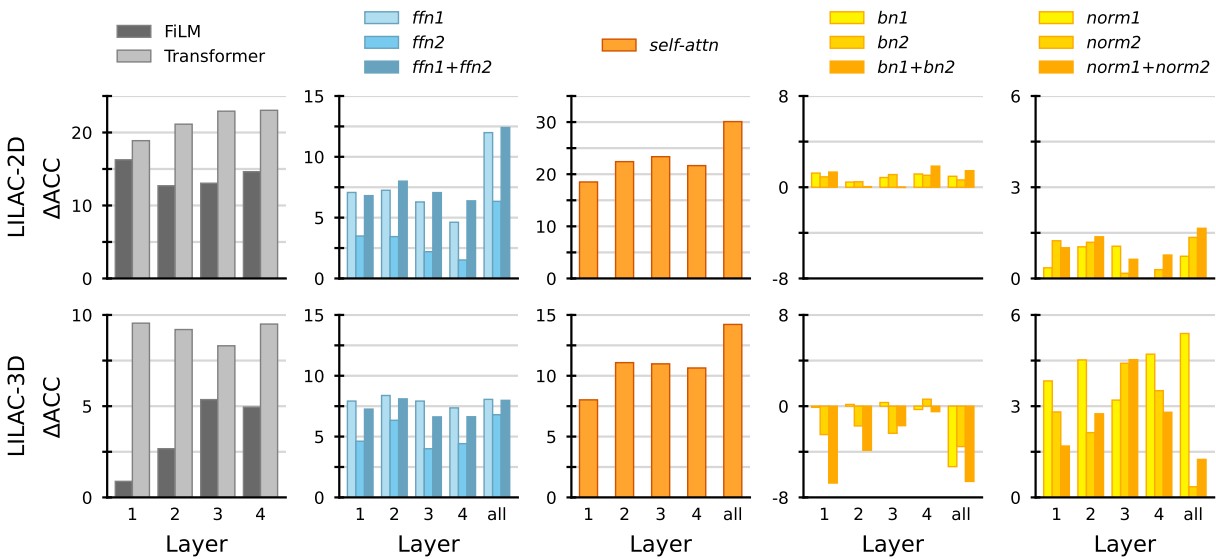

Figure 6: Evaluation of different selection strategies $\mathcal{S}$ trained with A&C for specialization. **1st column:** Layer-depth analysis. **2nd+3rd column:** Transformer feed-forward and multi-head attention modules. **4th+5th column:** Normalization scaling factors.

be identified and put into a shared context, we hypothesize that specialization on LILAC-2D tasks happens early and specialization on the LILAC-3D tasks happens late in the FiLMed network. As shown in the first column of Fig. 6, introducing task specialization to the first layer yields the highest accuracy gain for LILAC-2D, while the highest accuracy gain for LILAC-2D can be achieved by introducing specialized parameters to the penultimate or the last layer, which confirms our hypothesis.

**Transformer feed-forward and self-attention blocks.** Geva et al. (2021) consider the outputs of the feed-forward networks to be a composition of their key-value memories and discover that such layers learn some semantic patterns, especially in early layers. Consequently, such layers can be suitable candidates for specialization during CL. The results of our experiments can be found in the second column of Fig. 6. We make two key observations, which are roughly consistent across datasets: First, isolating the first transformer feed-forward layer alone (*ffn1*) is as effective as isolating the entire feed-forward block (*ffn1+ffn2*). Second, a selective specialization of this layer improves performance most if applied in the early layers (particularly, isolation at the second transformer encoder layer yields the best performance throughout). These observations show that, when chosen carefully, selection strategies can achieve high effectiveness with only a small amount of specialized parameters.

In a recent work on parameter-efficient transfer-learning methods, Smith et al. (2023) show the efficacy of adapting self-attention blocks in a vision transformer to downstream image classification tasks while keeping the remaining transformer parameters frozen. To assess whether their findings can be transferred to our CL setting, we conduct experiments with specialized self-attention modules and report our results in the third column of Fig. 6. We observe a substantial increase in accuracy, especially for specialization in intermediate transformer layers, albeit the greatest increase (~30% on LILAC-2D, ~14% on LILAC-3D) is achieved by specializing self-attention parameters across all layers. Although it is worth noting that self-attention modules account for about 73% of the parameters in each transformer layer, the results indicate a clear benefit from including self-attention parameters in specialization strategies.

**Normalization scaling factors.** Bilen and Vedaldi (2017) argue that specialized instance, layer, or batch normalization scaling factors can reduce task-specific biases, allowing them to intercept distributional shifts at low additional memory cost. However, as can be seen in the penultimate column of Fig. 6, specializing batch normalization parameters of the FiLMed network hardly improves (LILAC-2D) or even degrades (LILAC-3D) performance compared with fully shared batch normalization parameters. Nevertheless, introducing task-specific layer

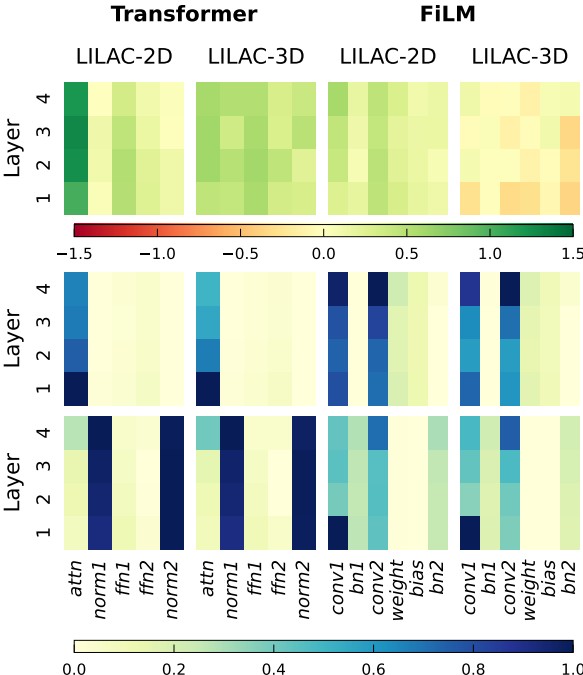

Figure 7: **1st row:** Accuracy gain of each specialized module $m \in \mathcal{M}$ over fine-tuning a monolithic network. Each field represents an experiment with $\mathcal{S} = \{m\}$. **2nd row:** Normalized gradient-based importance score $\text{IS}_{\text{grad}}(m)$. **3rd row:** Normalized activation-based importance score $\text{IS}_{\text{act}}(m)$.

normalization parameters in the VL transformer can slightly improve accuracy on the LILAC-2D tasks and achieves a performance gain of more than 5% on the LILAC-3D tasks. We additionally find that specializing layer normalization parameters that follow the multihead-attention operation (*norm1*) is generally more effective than specializing those that follow the feed-forward block (*norm2*), and even outperforms specializing both (*norm1+norm2*) on the LILAC-3D dataset.

### 4.2.3 Can we find quantifiable measures for the effectiveness of module selection strategies?

Quantifying the importance of neural connections to construct specialized weight masks is common practice in neural pruning research for CL (Molchanov et al., 2019). We aim to evaluate whether such importance scores can not only measure the suitability of specialization for single parameters but also for entire network modules. Therefore, we will compare two commonly used types of measurements for calculating the importance score (IS) of each module in the network: Gradient-based ($\text{IS}_{\text{grad}}$) and activation-based ($\text{IS}_{\text{act}}$) (cf. Wang et al., 2022; Gurbuz and Dovrolis, 2022;

Jung et al., 2020). Let $\theta^m$ denote the parameters of network module $m$ during training and let $\theta^m_t$ denote the parameters of the $m$-th module after training on the $t$-th task.

The gradient-based importance score $\text{IS}_{\text{grad}}(m)$ of a module is computed as the sum of the $L_1$-norm of each parameter $w$ of $m$ and the accumulated absolute gradients arriving at $w$ during training on each task:

$$\text{IS}_{\text{grad}}(m) := \alpha \sum_{t=1}^{T} \sum_{w \in \theta_m} |w| + \frac{1}{2} \left| \frac{\partial \mathcal{L}(\mathcal{D}_t; \theta)}{\partial w} \right|, \tag{2}$$

where the constant $\alpha = 1/(T \cdot \log(|\theta^m|))$ is used for averaging across all tasks and normalizing by the magnitude of the parameter count of $m$.

The activation-based importance score $\text{IS}_{\text{act}}(m)$ is computed as the total activation at module $m$ with the same normalization constant $\alpha$ as used above:

$$\text{IS}_{\text{act}}(m) := \alpha \sum_{t=1}^{T} \sum_{(\mathbf{l}_t, \mathbf{o}_t) \in \mathcal{D}_t} \left| f_{\theta^m_t}\left(g(\mathbf{l}_t), h(\mathbf{o}_t)\right) \right| \tag{3}$$

We calculate the Pearson coefficient (Pearson, 1895) between the importance scores $\text{IS}_{\text{grad}}(m)$ and $\text{IS}_{\text{act}}(m)$ of each network module $m \in \mathcal{M}$ (cf. Fig. 7, second and third row) and the relative accuracy gain yielded from isolating this module for task specialization ($\mathcal{S} = \{m\}$) (cf. Fig. 7, top row).

The Pearson values for the gradient-based importance score (FiLM/Transformer) indicate a strong positive correlation for the LILAC-2D tasks (0.91/0.90) and a weak positive correlation for the LILAC-3D tasks (0.09/0.40), respectively. Conversely, there is no conclusive evidence from the activation-based importance score (2D: 0.48/−0.53, 3D: −0.09/−0.32). Our results suggest that the magnitude of the gradients on the parameters of a module is a better indicator of the performance gain from specializing the whole module than the activation of the module during training. As shown in Fig. 7 (second row), the 2D convolutions in the last FiLMed layer and the attention modules in the VL transformer have high importance and thus seem to be particularly suited for specialization.

Table 1: ACC scores of baselines for the proposed datasets and model architectures. All selective specialization baselines are trained with A&C learning. Results from other selection strategies as well as forgetting and forward transfer measures can be found in Appendix B.

| | Transformer | | FiLM | |
|---|---|---|---|---|
| **Baseline** | 2D | 3D | 2D | 3D |
| Expert | 85.9 | 88.4 | 76.5 | 78.7 |
| MTL | 88.3 | 95.4 | 87.1 | 80.1 |
| SFT | 51.1 | 67.0 | 52.1 | 63.2 |
| ER | 52.4 | 77.7 | 53.1 | 66.9 |
| EWC | 55.5 | 79.0 | 56.1 | 69.0 |
| $\mathcal{S}_{\text{first-layer}}$ | 70.0 | 76.5 | 68.4 | 64.2 |
| $\mathcal{S}_{\text{last-layer}}$ | 74.2 | 76.5 | 66.8 | 68.2 |
| $\mathcal{S}_{\text{all-ffn1}}$ | 63.1 | 75.0 | - | - |
| $\mathcal{S}_{\text{conv-last-layer}}$ | - | - | 59.4 | 62.5 |
| $\mathcal{S}_{\text{all-attn}}$ | 81.2 | 81.2 | - | - |
| $\mathcal{S}_{\text{all-attn}}$ + ER | 85.7 | 88.5 | - | - |
| $\mathcal{S}_{\text{all-attn}}$ + EWC | 87.1 | 87.9 | - | - |

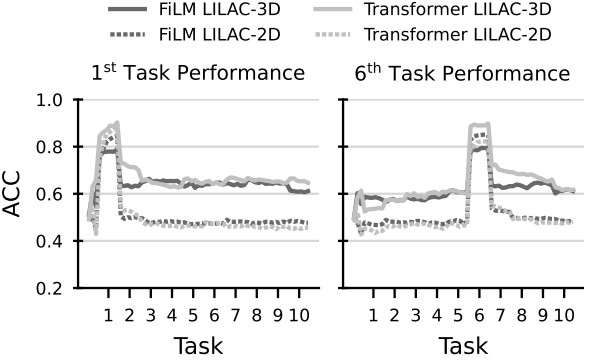

Figure 8: Accuracy of the SFT baseline on the first and the sixth task during training. LILAC-2D tasks are almost completely unlearned during training of the following tasks. Conversely, after training on LILAC-3D tasks, model performance has a slower decrease and tasks are not fully unlearned despite a lack of means to prevent forgetting.

### 4.2.4 How does the performance of different specialization strategies compare against CL baselines?

Based on the analyses conducted in Sec. 4.2.2 and Sec. 4.2.3, our aim is to determine whether selective task specialization can outperform CL baselines for the proposed LILAC datasets. Given that all possible selection strategies form a power set that grows exponentially with the number of modules in a network, we choose the following strategies for baseline comparison: In line with the findings on specialization at different layer depths, we compare with specialization of the first layer ($\mathcal{S}_{\text{first-layer}}$) and last layer ($\mathcal{S}_{\text{last-layer}}$) for transformer and FiLMed blocks, respectively. As we found the selection of the first feed-forward network in a transformer block to be particularly effective, we further compare with the specialization of such layer across all blocks ($\mathcal{S}_{\text{all-ffn1}}$). Finally, we add the two selection strategies of specializing self-attention modules in the transformer encoder ($\mathcal{S}_{\text{all-attn}}$) and the convolutions in the last FiLMed layer ($\mathcal{S}_{\text{conv-last-layer}}$) for comparison, as the corresponding modules yield the highest $\text{IS}_{\text{grad}}$ scores. The results are reported in Tab. 1.

Multiple selection strategies exhibit superior performance compared to CL baselines on LILAC-2D tasks. However, the sole strategy that surpasses all baselines on LILAC-3D is the specialization of attention across all layers. A possible explanation is that learning LILAC-2D tasks is generally

more brittle and subject to forgetting than learning LILAC-3D tasks, as can be seen in Fig. 8. Thus, a reasonable module specialization strategy significantly increases a model's robustness to forgetting.

An advantage of selective specialization is that it can be orthogonally combined with CL methods following the paradigms of replay or regularization. We find in our experiments that combining the most successful specialization strategy $\mathcal{S}_{\text{all-attn}}$ with ER or EWC by performing rehearsal or regularization during the consolidation phase of A&C learning reaches an accuracy close to the network of experts and even outperforms it by a $1.2\%$ margin ($\mathcal{S}_{\text{all-attn}}$ + EWC on LILAC-2D). Such results indicate that a combination of selective specialization with other CL methods is not only robust to forgetting but actually promotes transfer between shared modules, thus successfully striking the balance between specialization and generalization.

## 5 Related Work

**Continual learning.** In a CL scenario, an agent is exposed to a sequence of tasks with the objective of learning to solve the currently seen task while maintaining high performance on previous tasks. Approaches to CL can be broadly categorized into *regularization* (Kirkpatrick et al., 2017; Zenke et al., 2017; Li and Hoiem, 2018; Aljundi et al., 2018) that restrict parameter updates to bound plasticity, *(pseudo-)rehearsal* (Rebuffi et al., 2017; Chaudhry et al., 2019a; Buzzega et al., 2021), where data that are either retrieved from a memory buffer or synthetically generated from previous distributions are

periodically replayed to the model, and *dynamic architectures* (Rusu et al., 2016; Yoon et al., 2018), where models are gradually expanded in response to distributional shifts.

A few works explore the intersection of continual learning and visually grounded language learning with diagnostic (Skantze and Willemsen, 2022; Greco et al., 2019) and real-world (Srinivasan et al., 2022; Jin et al., 2020) datasets. All works conclude that common CL baselines struggle with striking a balance between forgetting and cross-task knowledge transfer, yet do not provide any insights on how this struggle is connected with the learning behaviors of the architecture used.

**Introducing task-specific parameters.** Research on continual neural pruning assigns some model capacity to each task by iteratively pruning and retraining (sub-)networks that are specialized to each task (Mallya and Lazebnik, 2018; Geng et al., 2021; Dekhovich et al., 2023; Kang et al., 2022; Hung et al., 2019; Gurbuz and Dovrolis, 2022; Jung et al., 2020; Wang et al., 2022). However, while such methods are effective in overcoming forgetting, the evolution and learning behavior of pruned subnetworks provides little interpretability regarding the role of individual parts of the network in solving CL problems.

Another line of research is modular CL, which trains a structural combination of independent parameterized components under the common assumption that each component solves a subproblem of each given task (Mendez and Eaton, 2021; Mendez et al., 2022; Ostapenko et al., 2021; Veniat et al., 2021). Similarly to neural pruning, approaches to modular CL fail to explain which role the interplay between learnable structural configurations and shared components takes in solving the tasks. In this topic area, the analysis provided by Csordás et al. (2021) is closely related to our work, except that we analyze model behavior on the level of entire modules rather than isolated parameters.

In an effort to promote parameter-efficient transfer in CL with foundation models, recent works utilize task-specific plugins such as capsule networks (Ke et al., 2021a) or adapters (Ke et al., 2021b; Zhao et al., 2022; Ermis et al., 2022) that leverage the pretrained representations as shared structure. This line of research is parallel to ours rather than competing, as we analyze existing parts of a trainable model rather than adding additional components to pretrained networks.

## 6 Conclusion

Striking a balance between specialization and generalization poses a challenge for agents that learn a sequence of language-conditioned tasks grounded in a visual environment. In this work, we consider vision-language models as a composition of modules and propose two datasets that allow us to analyze and compare strategies to selectively specialize modules to continual tasks with respect to different layer depths and module types. We further establish a gradient-based importance measure that quantifies the suitability of modules for specialization. Finally, we show that the module specialization strategies found in our analysis outperforms common CL baselines when trained under a conceptually simple adaptation-consolidation learning scheme. With this work, we show the merit of designing CL methods based on a careful analysis of selective specialization. Beyond introducing specialization to existing model parts, we plan to leverage our insights for the design of parameter isolation CL methods that introduce additional parameters into a pretrained model for future work.

## Limitations

Generally, we consider the results and analyses provided in this paper to be merely a cornerstone towards gaining more knowledge about model behavior during learning, an insight that can be used to better understand where in a model to introduce task-specific parameters. Nevertheless, we recognize the following limitations of our work: First, we conduct our experiments with two established vision-language architectures with just one model configuration each. More notably, the architectures we use are smaller and conceptually simpler than those used for large-scale realistic datasets on visual language grounding. Second, introducing task specialization to network modules naturally has a linear growth with the number of tasks. This can, however, be mitigated by choosing smaller modules for specialization with reasonable performance to strike a balance between parameter size and model performance.

## Acknowledgements

The authors gratefully acknowledge support from the DFG (CML, MoReSpace, LeCAREbot), BMWK (SIDIMO, VERIKAS), and the EU Commission (TRAIL, TERAIS).

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

# Appendices

## A  Experimental Settings

### A.1  LILAC-2D and LILAC-3D Datasets

Examples of both datasets are provided in Fig. 2 and Fig. 3. For the LILAC-2D dataset, object colors are {blue, green, grey, purple, red, yellow}, object types are {ball, box, key}, and directions are {down, to the left, to the right, up}. For the LILAC-3D dataset, blocks have the colors {blue, green, grey, purple, red, yellow}, bowls have the colors {brown, cyan, orange, petrol, pink, white}, and block sizes are {big, small}.

### A.2  Model Architectures

In the following, we describe further details about the two model architectures used in our experiments.

**FiLM.**  The FiLMed layer is illustrated in the left part of Fig. 4. We largely follow the architectural design of the FiLMed layers as suggested in Hui et al. (2020): Each FiLMed layer consists of a ConvBlock and the linear modulation fully-connected layers $\gamma$ (weight) and $\beta$ (bias) that condition the feature maps of the convolutional layers by scaling by the factor $\gamma$ and shifting by the value $\beta$. The ConvBlock consists of two two-dimensional convolutional layers with a kernel size of 3, a stride of 1, and a padding of 1, and two batch norm layers. Features of visual observations $\mathbf{o}_t$, $\mathbf{o}_t^+$, and $\mathbf{o}_t^-$ are initially extracted by the first block of a ResNet-18 (this configuration yielded better performance on both datasets than using the feature extractor as proposed in Hui et al. (2020)). Subsequently, the visual features of the premise observation $\mathbf{o}_t$ are passed through the first convolutional layer, followed by batch normalization, rectified linear unit (ReLU) activation, and the second convolutional layer. The language instruction $\mathbf{l}_t$ is passed through a word embedding layer, followed by a single-layer GRU. The embedded instruction sequence is passed through the two fully connected layers $\beta$ and $\gamma$, the output of which is used to modulate the visual features from the second convolutional layer. Finally, the modulated features are passed through a second batch normalization layer and ReLU activation. After passing $L$ FiLMed blocks, the features are max-pooled across the height and width of the feature maps and finally passed through a linear projection layer. In contrast to the 'premise' observation $\mathbf{o}_t$,

ResNet-18 features of the visual observations $\mathbf{o}_t^+$ and $\mathbf{o}_t^-$ are directly max-pooled and passed through the linear projection layer.

**Vision-Language Transformer.**  The right part of Fig. 4 provides an overview of the transformer architecture used in our experiments. Each transformer encoder layer follows the original design as proposed in Vaswani et al. (2017) and consists of a multi-head attention operation, two normalization layers, and two fully connected feed-forward layers. Similarly to the FiLM model, visual features of $\mathbf{o}_t$, $\mathbf{o}_t^+$, and $\mathbf{o}_t^-$ are extracted from the first block of a ResNet-18 encoder and passed through an additional linear encoder layer to match the latent dimension of the word embeddings from the input instruction $\mathbf{l}_t$. Word embeddings of $\mathbf{l}_t$ and visual features of $\mathbf{o}_t$ are then concatenated and fed to a multi-head attention layer, followed by a residual adding operation layer normalization. Afterwards, the latent VL features are passed through two fully connected linear layers, i.e., the feed-forward blocks of the network, which is again followed by a residual operation and layer normalization. After passing the total of $L$ transformer encoder layers, the features are mean-pooled across the height and width of the language-conditioned feature maps and fed to a linear projection layer. In the same way as with the FiLM architecture, visual features of $\mathbf{o}_t^+$ and $\mathbf{o}_t^-$ from the ResNet-18 are directly pooled and passed through the projection layer.

### A.3  Hyperparameter Configuration

An overview of the design choices and selected parameters for the model architectures can be found in Tab. 2.

### A.4  Metrics and Performance Evaluation

During training, we use the InfoNCE loss (Oord et al., 2019) $\mathcal{L}_{\text{info}}$ to maximize the cosine similarity between $d(f_{\theta_t}(h(\mathbf{o}_t), g(\mathbf{l}_t)))$ and $d(h(\mathbf{o}_t^+))$ and minimize the cosine similarity between $d(f_{\theta_t}(h(\mathbf{o}_t), g(\mathbf{l}_t)))$ and $d(h(\mathbf{o}_t^-))$. During inference, the model predicts the target image by choosing the image representation of $d(h(\mathbf{o}_t^+))$, $d(h(\mathbf{o}_t^-))$ that has higher cosine similarity with the representation of $d(f_{\theta_t}(h(\mathbf{o}_t), g(\mathbf{l}_t)))$, i.e.

$$S_{\cos(d(f_{\theta_t}(h(\mathbf{o}_t),g(\mathbf{l}_t))),d(h(\mathbf{o}_t^+))}$$
$$> S_{\cos(d(f_{\theta_t}(h(\mathbf{o}_t),g(\mathbf{l}_t))),d(h(\mathbf{o}_t^-)))} \quad (4)$$

Table 2: Overview of the hyperparameters selected for the model architectures with respect to the two datasets. We used the Weights&Biases (https://wandb.ai/site) sweep with Bayes optimizer.

| | Transformer | | FiLM | | Search space |
|---|---|---|---|---|---|
| | 2D | 3D | 2D | 3D | |
| init lr | 4.5e-4 | 2e-4 | 4.5e-4 | 2e-4 | [1e-6, 1e-3] |
| continual lr | 6e-4 | 7e-4 | 8e-4 | 1e-3 | [1e-6, 1e-3] |
| batch size | 128 | 128 | 128 | 128 | - |
| init epochs | 10 | 10 | 10 | 10 | - |
| continual epochs (per $t$) | 30 | 30 | 30 | 30 | - |
| word embedding dim | 256 | 256 | 128 | 128 | {32, 64, 128, 256, 512} |
| instr embedding dim | - | - | 256 | 256 | {32, 64, 128, 256, 512} |
| ResNet-18 layer features | 1 | 1 | 1 | 1 | {1, 2, 3, 4} |
| encoder layers ($L$) | 4 | 4 | 4 | 4 | { 1, 2, 4, 6, 8, 10, 12 } |
| attn heads | 2 | 2 | - | - | { 1, 2, 4, 6, 8} |
| ffn dim | 64 | 64 | - | - | {32, 64, 128, 256, 512} |
| EWC discount | 0.9 | 0.9 | 0.9 | 0.9 | [ 0, 1 ] |
| EWC $\lambda$ (joint) | 2,000 | 600 | 2,000 | 600 | [1e-2, 1e14] |
| EWC $\lambda$ (A&C) | 20,000 | 20,000 | 20,000 | 20,000 | [1e-2, 1e14] |
| ER buffer size | 3,000 | 3,000 | 3,000 | 3,000 | - |

The prediction accuracy $A_t$ can then be calculated as the number of samples of the $t$-th task for which Equation 4 holds, divided by the total number of samples from the $t$-th task, $|\mathcal{D}_t|$.

**Accuracy, transfer, forgetting.** We provide more details on evaluation metrics and some additional results on catastrophic forgetting (CF), and forward transfer (FT). Let $A_{i,j}$ denote the test accuracy of a model on the $j$-th task after observing the last sample of the $i$-th task. We largely follow the commonly used evaluation metrics as proposed in Lopez-Paz and Ranzato (2017):

$$\text{ACC} := \frac{1}{T}\sum_{t=1}^{T} A_{T,t}, \quad \text{CF} := \frac{1}{T}\sum_{t=1}^{T-1} A_{t,t} - A_{T,t}$$

$$\text{FT} := \frac{1}{T-1}\sum_{t=2}^{T} A_{t-1,t} - A_{\text{init},t}, \quad (5)$$

where $A_{\text{init},t}$ denotes the performance on the $t$-th task after initialization. Note that CF can be also interpreted as negative backward transfer, as it describes the influence of training on a task on previously seen tasks.

### A.5 Adaptation-Consolidation Algorithm

The pseudocode of the A&C training algorithm as a modification to the lifelong compositional learning algorithm proposed by Mendez and Eaton (2021)

is shown in Alg. 1. $\theta^{\mathcal{M}\setminus\mathcal{S}}$ denotes all shared parameters, whereas $\theta_t^{\mathcal{S}}$ are the task-specific, or *isolated*, parameters. We choose adaptFreq= 6 for our experiments.

---

**Algorithm 1** Adaptation-Consolidation (A&C)

Select modules $\mathcal{S} \subset \mathcal{M}$ for task specialization
Initialize all model parameters via joint training on $\mathcal{D}_{t_{\text{init}}}$
**for** $t = 1..T$ **do**
 Freeze $\theta^{\mathcal{M}\setminus\mathcal{S}}$
 **for** $e = 1..$adaptationEpochs **do**
  Update $\theta_t^{\mathcal{S}}$ upon training on $\mathcal{D}_t$
  **if** $e$ mod adaptFreq $= 0$ **then**
   Freeze $\theta_t^{\mathcal{S}}$, unfreeze $\theta^{\mathcal{M}\setminus\mathcal{S}}$
   Update $\theta^{\mathcal{M}\setminus\mathcal{S}}$ upon training on $\mathcal{D}_t$
   Freeze $\theta^{\mathcal{M}\setminus\mathcal{S}}$, unfreeze $\theta_t^{\mathcal{S}}$
  **end if**
 **end for**
**end for**

---

## B Additional Results

We provide some additional results on the effect of introducing task specialization to network modules in Fig. 9, Fig. 10, and Fig. 11. We provide more detailed results from our baseline comparison in Tab. 3. Finally, we provide two additional findings that might be interesting to the community:

**FiLM modulation parameters.** We observe that introducing task-specific parameters to either the weight ($\gamma$) or bias ($\beta$) separately yields the highest accuracy gain and forward transfer as well as the lowest forgetting. However, while the FiLMed model trained on the LILAC-2D tasks benefits most from weights to be specialized, the same model trained on LILAC-3D needs bias to be specialized to optimize performance metrics.

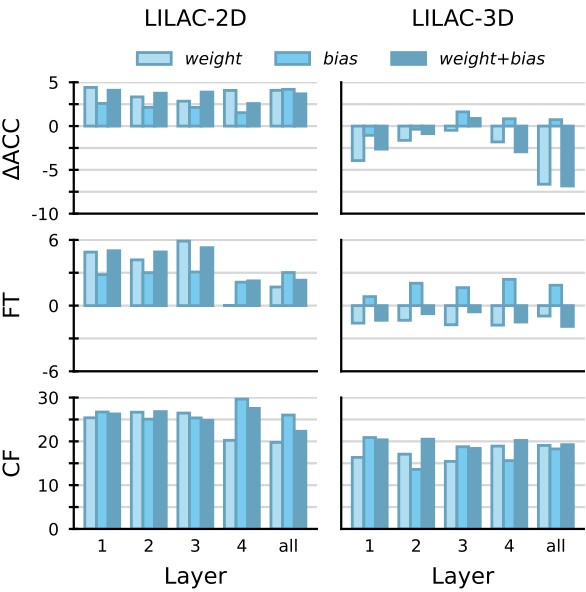

Figure 9: Analysis of specializing feature-wise linear modulation parameters in different layers. Top to bottom: 1) accuracy gain of specialization and A&C compared with monolithic SFT baseline, 2) forward transfer, and 3) forgetting.

**Modules whose specialization maximized forward transfer.** While for the FiLMed network the specialization of modulation parameters (weight for LILAC-2D, bias for LILAC-3D) maximizes the forward transfer, for the VL transformer it is the feed-forward layers and the layer normalization parameters that maximize forward transfer when specialized.

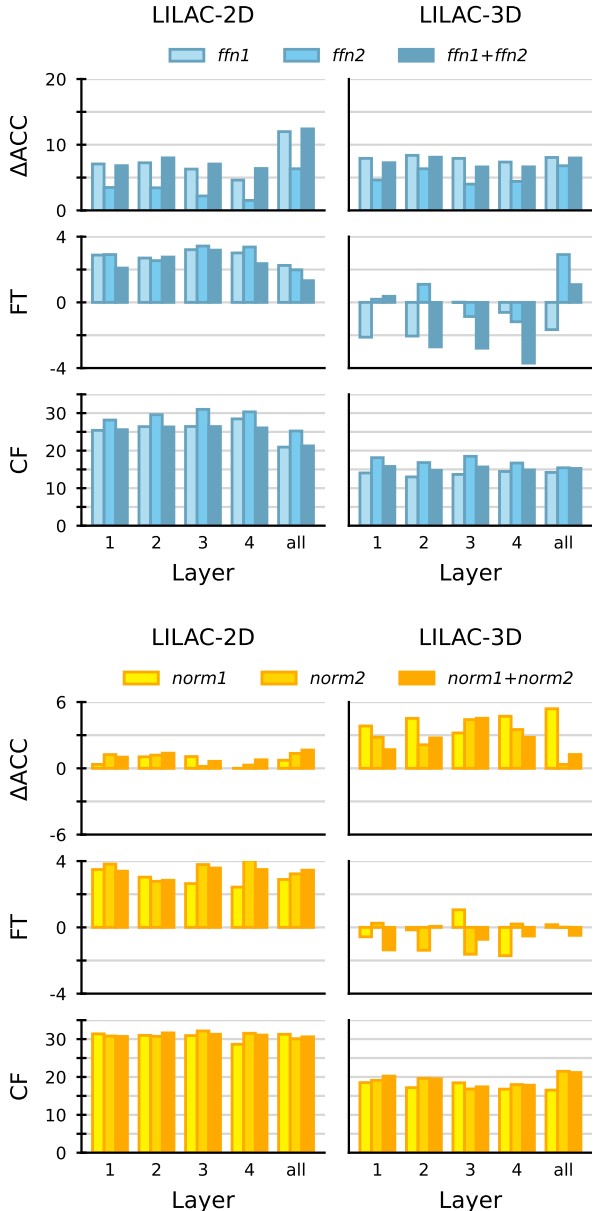

Figure 10: Analysis of specialization in different layers for feed-forward layers (top) and layer normalization parameters (bottom). Each plot shows (from top to bottom): 1) accuracy gain of specialization and A&C compared with monolithic SFT baseline, 2) forward transfer, and 3) forgetting.

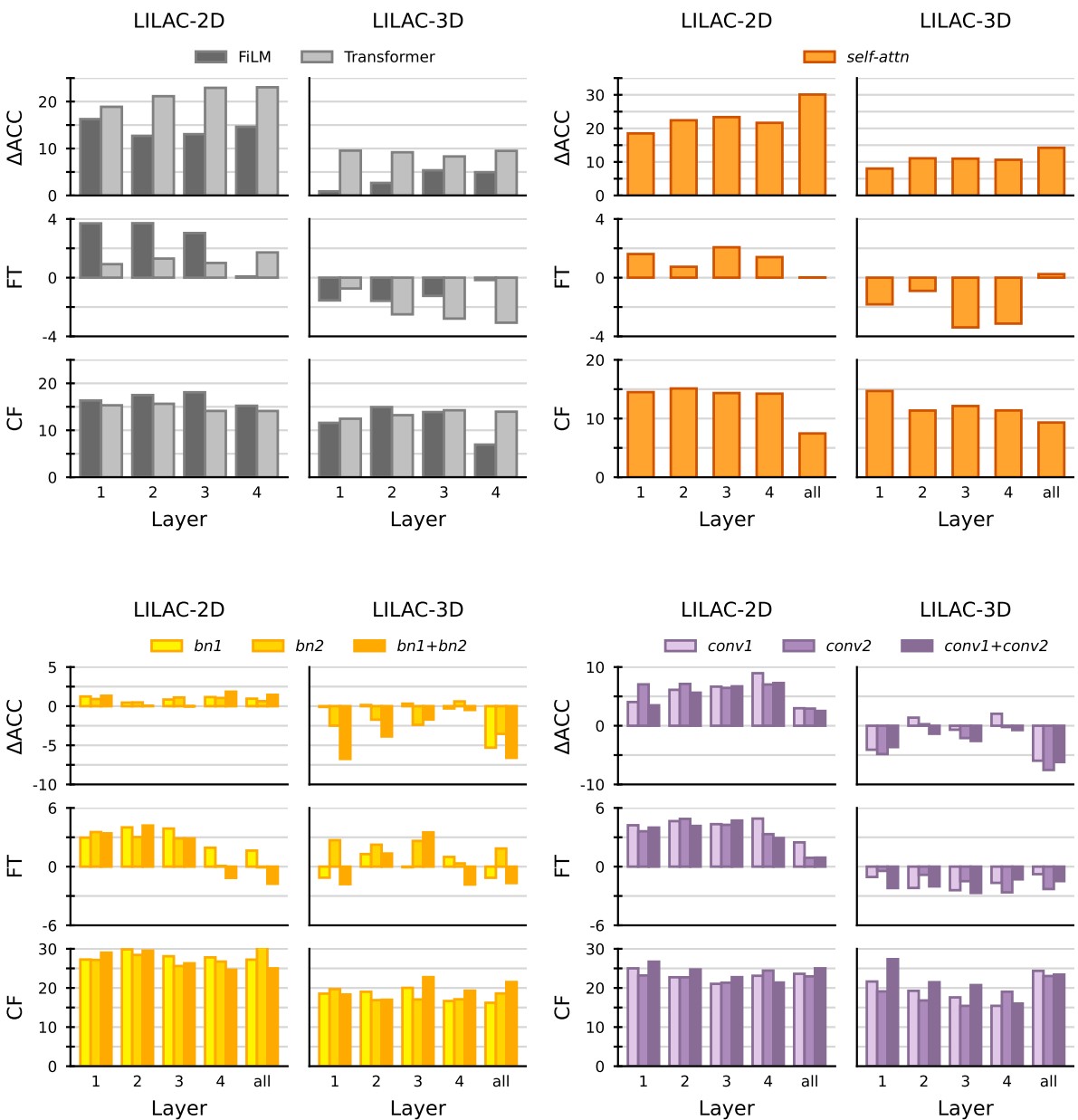

Figure 11: Analysis of specialization of layers in different depths (top left) and module specialization in different layers for self-attention (top right), batch normalization parameters (bottom left), and convolutional layers (bottom right). Each plot shows (from top to bottom): 1) accuracy gain of specialization and A&C compared with monolithic SFT baseline, 2) forward transfer, and 3) forgetting.

| FiLM | LILAC-2D | | | LILAC-3D | | |
|---|---|---|---|---|---|---|
| | ACC | FT | CF | ACC | FT | CF |
| MTL | $87.1 \pm 0.9\%$ | - | - | $80.1 \pm 1.1\%$ | - | - |
| Expert | $76.5 \pm 1.4\%$ | - | - | $78.7 \pm 2.7\%$ | - | - |
| SFT | $52.1 \pm 0.3\%$ | $2.4 \pm 0.4\%$ | $35.0 \pm 1.0\%$ | $63.3 \pm 0.8\%$ | $2.5 \pm 1.0\%$ | $17.5 \pm 1.7\%$ |
| ER | $53.1 \pm 0.3\%$ | $2.7 \pm 0.3\%$ | $33.9 \pm 0.9\%$ | $66.9 \pm 1.2\%$ | $4.8 \pm 1.4\%$ | $14.6 \pm 1.7\%$ |
| EWC | $56.1 \pm 0.9\%$ | $2.9 \pm 0.6\%$ | $7.7 \pm 0.9\%$ | $69.0 \pm 1.4\%$ | $3.6 \pm 0.6\%$ | $10.1 \pm 0.8\%$ |
| $\mathcal{M}_{\text{first-layer}}$ | $68.4 \pm 0.8\%$ | $3.7 \pm 0.4\%$ | $16.4 \pm 0.5\%$ | $64.2 \pm 2.1\%$ | $-1.6 \pm 0.5\%$ | $11.6 \pm 0.9\%$ |
| $\mathcal{M}_{\text{last-layer}}$ | $66.8 \pm 0.8\%$ | $0.1 \pm 0.1\%$ | $15.2 \pm 0.4\%$ | $68.2 \pm 1.8\%$ | $-0.2 \pm 0.3\%$ | $7.0 \pm 0.9\%$ |
| $\mathcal{M}_{\text{conv-last-layer}}$ | $59.4 \pm 0.4\%$ | $2.9 \pm 0.5\%$ | $21.4 \pm 1.7\%$ | $62.5 \pm 0.9\%$ | $-1.3 \pm 0.6\%$ | $16.0 \pm 2.6\%$ |

| Transformer | LILAC-2D | | | LILAC-3D | | |
|---|---|---|---|---|---|---|
| | ACC | FT | CF | ACC | FT | CF |
| MTL | $88.3 \pm 0.3\%$ | - | - | $95.4 \pm 0.3\%$ | - | - |
| Expert | $85.9 \pm 0.4\%$ | - | - | $88.4 \pm 0.2\%$ | - | - |
| SFT | $51.1 \pm 0.2\%$ | $2.5 \pm 0.6\%$ | $35.7 \pm 0.6\%$ | $67.0 \pm 0.8\%$ | $-2.5 \pm 0.8\%$ | $24.2 \pm 0.6\%$ |
| ER | $52.4 \pm 0.4\%$ | $2.7 \pm 0.3\%$ | $34.6 \pm 0.5\%$ | $77.7 \pm 0.8\%$ | $1.6 \pm 0.9\%$ | $12.7 \pm 0.7\%$ |
| EWC | $55.5 \pm 0.7\%$ | $2.2 \pm 0.6\%$ | $14.2 \pm 0.4\%$ | $79.0 \pm 0.7\%$ | $1.8 \pm 0.9\%$ | $8.9 \pm 0.6\%$ |
| $\mathcal{M}_{\text{first-layer}}$ | $70.0 \pm 0.4\%$ | $0.9 \pm 0.3\%$ | $15.3 \pm 0.6\%$ | $76.5 \pm 0.7\%$ | $-0.7 \pm 0.4\%$ | $12.5 \pm 0.7\%$ |
| $\mathcal{M}_{\text{last-layer}}$ | $74.2 \pm 0.6\%$ | $1.7 \pm 0.3\%$ | $14.1 \pm 0.4\%$ | $76.5 \pm 1.0\%$ | $-3.1 \pm 0.6\%$ | $14.0 \pm 1.0\%$ |
| $\mathcal{M}_{\text{all-ffn1}}$ | $63.1 \pm 0.5\%$ | $2.3 \pm 0.3\%$ | $20.9 \pm 0.5\%$ | $75.0 \pm 0.7\%$ | $-1.7 \pm 0.9\%$ | $14.2 \pm 1.0\%$ |
| $\mathcal{M}_{\text{all-attn}}$ | $81.2 \pm 0.6\%$ | $0.0 \pm 0.3\%$ | $7.5 \pm 0.7\%$ | $81.2 \pm 0.8\%$ | $0.2 \pm 0.5\%$ | $9.3 \pm 0.9\%$ |
| $\mathcal{M}_{\text{all-attn}}$ + ER | $85.7 \pm 1.7\%$ | $0.1 \pm 0.2\%$ | $0.8 \pm 0.2\%$ | $88.5 \pm 0.5\%$ | $-0.8 \pm 0.2\%$ | $0.4 \pm 0.2\%$ |
| $\mathcal{M}_{\text{all-attn}}$ + EWC | $87.1 \pm 0.4\%$ | $0.4 \pm 0.1\%$ | $0.6 \pm 0.2\%$ | $87.9 \pm 0.5\%$ | $-0.5 \pm 0.3\%$ | $0.6 \pm 0.1\%$ |

Table 3: Average accuracy (ACC), forward transfer (FT), and forgetting (CF) of FiLM (top) and vision-language transformer (bottom) baselines for all datasets. Results are averaged across ten seeds. Standard error after $\pm$.