# OpenReview forum: "Visually Grounded Continual Language Learning with Selective Specialization"
_EMNLP/2023/Conference — EMNLP 2023 Findings_

### Official Review · Reviewer_LwBz · 2023-08-04

**Soundness:** 3

**Excitement:**

2: Mediocre: This paper makes marginal contributions (vs non-contemporaneous work), so I would rather not see it in the conference.

**Paper Topic And Main Contributions:**

The paper propose two datasets related to  visual grounded continual learning and  applies A&C algorithm on these two datasets to study the relation between specialization and generalization.

**Questions For The Authors:**

- How are the proposed two datasets related to continual learning? Because we only the each item one time?
- The problem in the  proposed dataset are not classification problem (select one image based on current image and instruction)? if it is a classification problem, why the paper talk the limitation in line 154-157.
- what does grounding mean in this work? Just consider the visual input information?

**Reasons To Accept:**

- Apply A&C in visual grounded setting.
- Propose two simple datasets.

**Reasons To Reject:**

- Writing needs significant improvement.
- The proposed method is incremental. This paper just propose two simple datasets and apply A&C methods on two base models to check the performance in these two datasets.

**Reproducibility:**

3: Could reproduce the results with some difficulty. The settings of parameters are underspecified or subjectively determined; the training/evaluation data are not widely available.

**Reviewer Confidence:**

3: Pretty sure, but there's a chance I missed something. Although I have a good feel for this area in general, I did not carefully check the paper's details, e.g., the math, experimental design, or novelty.

**Typos Grammar Style And Presentation Improvements:**

- line 82 to 86, quite confusing
- line 178

---

> ### Author Rebuttal · Authors · 2023-08-29
>
> We thank $\textcolor{purple}{\textsf{LwBz}}$ for reviewing our paper.
>
> ## 1) Writing Quality
>
> **Q1:**
> > Writing needs significant improvement.
>
> **A1:** To better address this concern, we kindly request more specific information regarding the areas of the paper that require improvement in writing. Any **specific examples or suggestions** would be incredibly helpful as we work to enhance the quality of the paper.
>
>
> ## 2) Contributions
>
> **Q2:**
> > This paper just propose two simple datasets and apply A&C methods on two base models to check the performance in these two datasets.
>
> **A2:** As was correctly noticed by the other reviewers ($\textcolor{blue}{\textsf{Lw5h}}$, $\textcolor{orange}{\textsf{FZU5}}$, $\textcolor{teal}{\textsf{F5yK}}$) and mentioned in our paper (e.g., lines 41-49), our aim is to **study different strategies for selective specialization** to **understand the role of each network module** during continual learning of language grounded in visual observations. Consequently, this paper is considered an analysis paper rather than a methodology paper.
>
> ## 3) Other Questions
>
> **Q3.1:** Reviewer $\textcolor{purple}{\textsf{LwBz}}$ asks whether the scenario proposed with the datasets can be considered a classification problem and asks for clarification why training task-specific output heads as a specialization strategy is not applicable in our work.
>
> **A3.1:** Although the learning setting could be reformulated as a binary classification problem, we designed a scenario inspired by **contrastive supervised learning for Natural Language Inference** (cf. lines 122-125). Our model therefore learns contrasting latent representations, but does not produce any logits for multi-class classification. As for our analysis of selective specialization strategies, this **prevents a trivial solution of learning task-specific output heads**, as done in prior literature.
>
> **Q3.2:**
> > How are the proposed two datasets related to continual learning? Because we only the each item one time?
>
> **A3.2:** The proposed datasets indeed align with the continual learning (CL) paradigm, as they reflect **scenarios involving sequential and non-stationary data**. Contrary to the assumption, **each item is encountered multiple times**, as the model is trained on each task for multiple epochs (cf. Table 2). However, **training once on each data point is not considered a necessary criterion for a generic CL scenario**, but relates to the specific subfield of Online Continual Learning, which we do not address in this work.
>
> **Q3.3:**
> > What does grounding mean in this work? Just consider the visual input information?
>
> **A3.3:** *Grounding* or *visual language grounding* (line 13) in our paper refers to the challenge of a model to establish a meaningful connection between visual and language inputs.
>
> In general, but notably also in the context of our work, grounding entails more than solely considering visual input information: The proposed LILAC 2D and 3D tasks can only be solved by **successfully understanding the premise scene** and **infering the next possible scene based on the language instruction**.

---

### Official Review · Reviewer_FZU5 · 2023-08-05

**Soundness:** 3

**Excitement:**

3: Ambivalent: It has merits (e.g., it reports state-of-the-art results, the idea is nice), but there are key weaknesses (e.g., it describes incremental work), and it can significantly benefit from another round of revision. However, I won't object to accepting it if my co-reviewers champion it.

**Paper Topic And Main Contributions:**

The paper focuses on creating artificial agents that excel in a sequence of language-informed tasks within the visual world. They propose new diagnostic datasets for continual learning in visual language grounding, comparing strategies for module specialization. Experimental results demonstrate the superiority of their strategies over some common baseline methods.

**Reasons To Accept:**

- the paper proposes two continual learning benchmarks in visual grounding setups.
- analysis on generalization vs. specialization in the paper.
- the author provided code in the supplementary material, which shows their reproducibility

**Reasons To Reject:**

First, the novelty is limited. Specifically
- why do we need these two datasets? what's the motivation of creating these two datasets? what's missing in the field that these datasets fill in? assuming this is a dataset paper, the author should talk more about this.
- what's the takeaway of the paper? assuming this is a method paper, is the message like (1) some module specialization helps on the continual learning and (2) the best specialization is on the self-attention layers? If so, I'd like to know the extra parameters required for self-attention specialization vs . LoRA or other adapter methods, otherwise I think the conclusion is not convincing enough.

Second, some baselines are not compared.
- EWC and ER are not the best regularization-based and rehearsal-based continual learning methods. The paper could compare with at least one more recent method to show the specialization argument is also valid on that.
- As the paper is more on the side of isolation-based methods, where the number of parameters grows with the number of tasks. It's better to compare with some method like progressive neural networks.

Third, the writing is hard to follow, especially the notation.
- Is mathcal M defined? Is it equal to the network in 4.1
- m is defined in Sec 2 but first used in the experiment section 4.2
- Until 4.2.4 I understood the meaning of "module" and it is not the same meaning as in the neural module network. Why not just call it a set of layers? Following the definition in Sec 2, the subtitle "best strategy" in 4.2 overclaims, the author didn't explore all the strategies, like pruning the network for a subset of parameters, etc.

**Reproducibility:**

5: Could easily reproduce the results.

**Reviewer Confidence:**

4: Quite sure. I tried to check the important points carefully. It's unlikely, though conceivable, that I missed something that should affect my ratings.

---

> ### Author Rebuttal · Authors · 2023-08-29
>
> We thank reviewer $\textcolor{orange}{\textsf{FZU5}}$ for the constructive feedback.
>
> ## 1) Dataset Novelty and Takeaway
>
> **Q1.1:** Reviewer $\textcolor{orange}{\textsf{FZU5}}$ asks to elaborate on the research gap that the two proposed datasets fill in.
>
> **A1.1:** We propose two novel LILAC datasets in our work due to the lack of benchmarks that could fulfill the following criteria in our visually grounded continual language learning setting (cf. lines 53-62):
>
> 1. High overlap between tasks
> 2. Compositional nature of a dataset
> 3. Comparable inter-task and intra-task similarity
> 4. Non-trivial task (i.e., high similarity between positive and negative samples to be contrasted)
> 5. Simulated environment instead of real-world images
>
> Criteria 1, 2, and 3 aim at **maximizing knowledge transfer** to bias a learner towards keeping many modules shared across tasks for generalization while *selectively* specializing a limited set of modules.
> Criteria 4 and 5 aim at **preventing the model from doing reasoning shortcuts** that might hinder the generalizability derived from findings about model behavior (nonetheless, we do consider bridging the gap between controlled environments and real-world setting as valuable future work).
>
> As the two proposed datasets fulfill the abovementioned criteria, they allow for a thorough analysis and provide insights insights into the continual learning (CL) behavior under different specialization strategies.
>
> **Q1.2:** Reviewer $\textcolor{orange}{\textsf{FZU5}}$ asks to elaborate on the takeaway and conclusions drawn from this work, particularly under the assumption that this is a methodology paper.
>
> **A1.2:** We consider this paper as an **analysis paper**. Particularly, our work has two main takeaways:
>
> Carefully balancing the trade-off between generalization and specialization
> 1. provides **insights into the behavior of individual model parts** (Sections 4.2.2 and 4.2.3). This could benefit the design of future parameter isolation CL methods, e.g., by answering where in the network to insert task-specific parameters;
> 2. helps us **design conceptually simple approaches that outperform common CL baselines** (Sections 4.2.1 and 4.2.4).
>
>
> As mentioned in lines 600-612, we consider adapter methods for pretrained models as a parallel rather than competing line of research (e.g., capsule networks, adapters).
>
> ## 2) Comparison with Additional Baselines
>
> **Q2:** Reviewer $\textcolor{orange}{\textsf{FZU5}}$ asks for 1) a comparison with a more recent CL baseline and 2) a comparison with a isolation-based method such as Progressive Neural Networks.
>
> **A2:** With reference to the two main takeaways described in **A1.2**, the **aim of this analysis paper is *not* to provide a new SOTA parameter isolation method**, but rather show the merit of careful selection of model parts to be specialized for CL.
>
> Nevertheless, we follow $\textcolor{orange}{\textsf{FZU5}}$'s request to conduct additional experiments to compare with a strong recent pruning method. Winning SubNetworks (WSN) [1] is a recent parameter-isolation method focused on learning binary task-wise masks for each parameter (*not* module) and selectively freezing parameters used in previous tasks. As it is designed for convolutional neural networks, we perform experiments on the FiLMed network only.
>
> Running each experiment with ten different random seeds and for two different task capacity ratios $c=0.1$ and $c=0.5$, we get the following results: WSN ($c=0.1$): $80.0 \pm 5.7\%$ (2D) / $78.9 \pm 7.4\%$ (3D), WSN ($c=0.5$): $81.4 \pm 2.2 \%$ (2D) / $77.8 \pm 9.3\%$ (3D).
>
> WSN achieves comparable performance with the self-attention specialization strategy for the Transformer backbone on the 2D and 3D datasets ($81.2\%$). This is, however, not surprising, as we claim in our paper that pruning methods are an effective and performant method for continual learning (lines 582-584). However, as we also claim in the paper (lines 584-587), they do not add interpretability and valuable insights in the role of individual model parts in learning to solve problems continually (cf., our first takeaway).
>
> To explain why we do not compare with Progressive Neural Networks [2] as suggested by reviewer $\textcolor{orange}{\textsf{FZU5}}$: This method adds a "column" every time the model sees a new task. In our case, this would imply to have a new copy of all modules for every task, which is in stark contrast to the objective of this paper to determine how to selectively specialize individual modules.
>
>
> ## 3) Terminology and Notation
>
> **Q3.1:** Reviewer $\textcolor{orange}{\textsf{FZU5}}$ asks for the definition of $\mathcal{M}$.
>
> **A3.1:** As mentioned in line 161, $\mathcal{M}$ is a set of modules (e.g., self-attention, FFNs, batch norm) that comprise a network $f$. We will make the definition more explicit.
>
> **Q3.2:** Reviewer $\textcolor{orange}{\textsf{FZU5}}$ asks to clarify on the meaning of *modules* $m$, especially with respect to Neural Module Networks (NMN), and asks whether $m$ corresponds to a set of layers.
>
> **A3.2:** In our paper, a *module* $m$ is any type of self-contained parametric function within a deep neural network that can be either entirely specialized or shared across tasks. Therefore, the meaning of module is not the same meaning as in the NMN (our architectures Transformers and FiLM networks are obviously not NMNs). We agree, however, that the term *module* can confuse the redears familiar with NMNs and will improve our description in Section 2.1 in that regard.
>
> We deliberately avoid the term *layer* in the introduction to not confuse it with, e.g., transformer encoder layers, which would consist of multiple modules like self-attention, layer norm etc.
>
> **Q3.3:** According to $\textcolor{orange}{\textsf{FZU5}}$, the term "best strategy" in Section 4.2 could be less assertive, as specialization strategies like parameter-specific network pruning were not explored in the analysis.
>
> **A3.3:** Our aim is to analyze selection strategies of *modules* (i.e., any type of self-contained parametric function within a deep neural network) and not *parameters*. While methods for parameter pruning have shown to be effective in preventing forgetting, they provide little interpretability regarding the role of individual model parts in solving CL problems (cf. 582-587). Hence, the scope of strategies for analysis is limited to modules.
>
> [1] Kang et al. "Forget-free Continual Learning with Winning Subnetworks" ICML 2022
> [2] Rusu et al. "Progressive Neural Networks" NIPS 2016 Deep Learning Symposium

---

### Official Review · Reviewer_F5yK · 2023-08-10

**Soundness:** 4

**Excitement:**

4: Strong: This paper deepens the understanding of some phenomenon or lowers the barriers to an existing research direction.

**Paper Topic And Main Contributions:**

The paper studies module specialization for continual learning in vision-and-language contrastive learning. To explore the trade-off between parameter specialization and knowledge transfer, the paper introduces the Lifelong Language Compositions (LILAC) benchmark that consists of two synthetic datasets with compositional instructions.

The paper utilizes an alternating learning scheme of adaptation and consolidation phases and tests different module selection strategies for two vision-and-language architectures (FiLM and cross-modal transformer). The experiments show that gradient-based importance scores can be used successfully for selecting modules for specialization and that the examined method can outperform standard baselines of ER and EWC.

**Reasons To Accept:**

- The paper studies selective specialization in vision-and-language models which is an interesting approach to continual learning.
- It introduces two datasets with compositional structure.
- The experiments explore various module selection strategies and show that the best strategy outperforms common baselines.
- The paper is written clearly and the findings are connected to the wider literature.

**Reasons To Reject:**

- Based on Table 1, specializing all self-attention modules seems to be by far the best strategy for transformer models. However, self-attention layers are not considered in the experiments for the previous research questions (sections 4.2.2 and 4.2.3). Since results per layer are presented in Figure 8 (Appendix B), it would be useful to include these results in the correlation analysis with the selection based on importance scores.
- The synthetic nature of the images and instructions in the dataset makes it unclear whether results in this benchmark will generalize in real-world settings where tasks would have more complex structure.


**Reproducibility:**

4: Could mostly reproduce the results, but there may be some variation because of sample variance or minor variations in their interpretation of the protocol or method.

**Reviewer Confidence:**

3: Pretty sure, but there's a chance I missed something. Although I have a good feel for this area in general, I did not carefully check the paper's details, e.g., the math, experimental design, or novelty.

**Typos Grammar Style And Presentation Improvements:**

- In Figure 2a, the negative values of the colorbar are not readable.
- In Figure 4, it is not clear what each column of the heatmap represents.

---

> ### Author Rebuttal · Authors · 2023-08-29
>
> We thank the reviewer $\textcolor{teal}{\textsf{F5yK}}$ for the detailed and constructive feedback and the positive assessment of our study.
>
> ## 1) Experimental Findings on Self-Attention Layer Specialization
>
> **Q1:** Reviewer $\textcolor{teal}{\textsf{F5yK}}$ asks to integrate the results on the self-attention layers as presented in Figure 8 in the correlation analysis.
>
> **A1:** This is already **done in the correlation analysis**, as we treat each specialized module in the transformer network (notably including self-attention layers) as a separate datapoint ($x_1$, $x_2$) where $x_1$ is the relative accuracy gain from specializing the module under the A&C learning scheme (cf. Figure 2 first row) and $x_2$ is the gradient-/activation-based importance score on this module (cf. Figure 4). However, we acknowledge that the connections between Figures 2 and 4 and drawn conclusions in the results subsections were not clear enough in the first version of this script, a remark also made by reviewer $\textcolor{blue}{\textsf{Lw5h}}$. Hence, we will **enhance the comprehensiveness of our evaluation** by expanding the depth of plot interpretation in Section 4.2.
>
> ## 2) Generalizability to Real-World Settings
>
> **Q2:** $\textcolor{teal}{\textsf{F5yK}}$ raises the question whether the results from the synthetic datasets will generalize to real-world setting with more complex tasks.
>
>
> **A2:** On the one hand, prior scenarios on continual vision-language grounding that are based on synthetic/diagnostic datasets are **too simplistic** either in terms of the continual learning problem (only one distributional shift) or in terms of the vision-language reasoning problem (simplistic images with one object, cf. lines 57-62). On the other hand, analyzing model behavior based real-world images may be **strongly biased by the visual reasoning shortcuts** that have been observed in models (cf. lines 67-74).
>
> To leverage the advantages of synthetic and real image scenarios, we provide two datasets that allow for
> 1) **investigating the learning behavior of the models with a high degree of control and flexibility** while
> 2) being challenging enough to require **object localization, spatial reasoning, concept learning,** and finally **language grounding** capabilities of the continual learner.
>
> We believe that by gaining promising insights with the benchmarks proposed in this paper, we lay the groundwork for further research that can bridge the gap between synthetic and real-world scenarios.
>
> ## 3) Minor Remarks on Formatting
>
> We thank reviewer $\textcolor{teal}{\textsf{F5yK}}$ for notifying us for some formatting and reference issues of the Figures 2a and 4 in the manuscript and will fix these in the revised version.

---

### Official Review · Reviewer_Lw5h · 2023-08-10

**Soundness:** 3

**Excitement:**

3: Ambivalent: It has merits (e.g., it reports state-of-the-art results, the idea is nice), but there are key weaknesses (e.g., it describes incremental work), and it can significantly benefit from another round of revision. However, I won't object to accepting it if my co-reviewers champion it.

**Paper Topic And Main Contributions:**

This paper studies learning new skills from language instructions in a continual learning setting. It formulates the problem as a selective parameter specialization problem which identifies parts/modules for parameter sharing or specialization.  It created two datasets and conducted experiments to study the tradeoff between generalization and specialization.  Experimental results are reported.

**Questions For The Authors:**

-	What’s the motivation of having 2D and 3D datasets? Any new challenges brought by 3D?
-	line 230-231,  why only 10 tasks (not 60 tasks?) need to be learned incrementally?


**Reasons To Accept:**

-	A well-motivated and scoped study to address the challenges with continual learning
-	The definition of the problem of selection strategy is intuitive and well-explained.
-	Two datasets to support research on this topic.


**Reasons To Reject:**

-	The result section seems to be written in a rush. Some parts are not straightforward to follow, e.g., how to interpret Figure 2?  The connections between the results (figures) and the general conclusions in the introduction section need to be better explained.
-	Since this is about continual learning, I would expect some results/analysis that show some sort of metrics along the time scale (e.g., X-axis).


**Reproducibility:**

3: Could reproduce the results with some difficulty. The settings of parameters are underspecified or subjectively determined; the training/evaluation data are not widely available.

**Reviewer Confidence:**

4: Quite sure. I tried to check the important points carefully. It's unlikely, though conceivable, that I missed something that should affect my ratings.

---

> ### Author Rebuttal · Authors · 2023-08-29
>
> We appreciate the reviewer $\textcolor{blue}{\textsf{Lw5h}}$'s constructive feedback on our paper and their positive assessment of the motivation, problem definition, and scope of our study.
>
>
> ## 1) Writing Quality in Sec. 4.2
>
> **Q1:** The reviewer $\textcolor{blue}{\textsf{Lw5h}}$ suggests improvements in the result section to enhance clarity, particularly in explaining Figure 2. Furthermore, the reviewer recommends strengthening the connections between the results and the conclusions drawn in the introduction.
>
> **A1:** We acknowledge your concern and will **provide more explicit explanations** for the interpretation of Figure 2 as well as other figures, and also **align our experimental results with the summarized conclusions** in the introduction.
>
>
> ## 2) Plot Metrics along Time Scale
>
> **Q2:** The reviewer $\textcolor{blue}{\textsf{Lw5h}}$ suggests to add plots for analysis/results that show the metrics along a time scale.
>
> **A2:** We agree that presenting results with a time-based perspective can provide deeper insights into the dynamics of continual learning. To address this concern, we will incorporate **plots of evaluation accuracy over time**, one plot for each dataset, **contrasting both the baselines and our specialized strategies utilized in the final comparison experiments** (Sec. 4.2.4).
>
> ## 3) Motivation for Dataset and Continual Stream Design
>
> **Q3.1:** Reviewer $\textcolor{blue}{\textsf{Lw5h}}$ asks for the motivation to propose the 3D dataset in addition to the 2D dataset.
>
> **A3.1:** While experiments with the 2D dataset provide valuable insights on the behavior of model parts during continual learning, we enrich our set of benchmarks with 3D data to **compare whether our findings still hold when increasing spatial complexity** and **introducing distracting information** such as the robot arm position. We further consider the introduction of 3D synthetic images as a step to bridge the gap to real 3D images while maintaining the control and flexibility for model analysis provided with a simulated benchmark.
>
> **Q3.2:** Reviewer $\textcolor{blue}{\textsf{Lw5h}}$ asks for the reason to choose 10 tasks over 60 tasks for the incremental learning setup.
>
> **A3.2:** Grouping multiple different instructions into a task (in our case, 10 tasks à 6 instructions, cf. lines 226-227) is a common practice in continual learning research. To give some examples, we refer to some of the most widely known continual learning benchmarks CIFAR-100 [1] (10 tasks à 10 classes), Permuted MNIST [2] (5 tasks à 2 digits) and CORe50 [3] (5 tasks à 20 object categories). As our extension of the contrastive learning problem in Natural Language Inference to visually grounded language is novel in our work, we aimed to **align with commonly agreed methods when designing the continual stream**.
>
> [1] Rebuffi et al. "iCaRL: Incremental Classifier and Representation Learning" CVPR 2017
> [2] Kirkpatrick et al. "Overcoming catastrophic forgetting in neural networks" PNAS 2017
> [3] Lomonaco & Maltoni "CORe50: a New Dataset and Benchmark for Continuous Object Recognition" CoRL 2017

---

### Meta-Review · Area_Chair_tMJQ · 2023-09-19

**Recommendation:** 3

**Metareview:**

Positives of the paper include a well motivated problem and novel datasets being produced. Improvements in the writing could make the paper stronger, particularly in the results section and the notation. Reviewers also expressed desire to see results expressed in terms of time being spent learning. It is also not clear whether the results would generalize to more realistic datasets and scenarios.

---

### Decision · Program_Chairs · 2023-10-07

**Decision:**

Accept-Findings

**Comment:**

Positives of the paper include a well motivated problem and novel datasets being produced. Improvements in the writing could make the paper stronger, particularly in the results section and the notation. Reviewers also expressed desire to see results expressed in terms of time being spent learning. It is also not clear whether the results would generalize to more realistic datasets and scenarios.